# Photocatalytic ethylene production by oxidative dehydrogenation of ethane with dioxygen on ZnO-supported PdZn intermetallic nanoparticles

Pu Wang[1,2], Xingyu Zhang[2,3], Run Shi[1] ✉, Jiaqi Zhao[1,2], Geoffrey I. N. Waterhouse[4], Junwang Tang[5,6] & Tierui Zhang[1,2] ✉

The selective oxidative dehydrogenation of ethane (ODHE) is attracting increasing attention as a method for ethylene production. Typically, thermo-catalysts operating at high temperatures are needed for C–H activation in ethane. In this study, we describe a low temperature (< 140 °C) photocatalytic route for ODHE, using $O_2$ as the oxidant. A photocatalyst containing PdZn intermetallic nanoparticles supported on ZnO is prepared, affording an ethylene production rate of 46.4 mmol $g^{-1}$ $h^{-1}$ with 92.6% ethylene selectivity under 365 nm irradiation. When we employ a simulated shale gas feed, the photo-catalytic ODHE system achieves nearly 20% ethane conversion while maintaining an ethylene selectivity of about 87%. The robust interface between the PdZn intermetallic nanoparticles and ZnO support plays a crucial role in ethane activation through a photo-assisted Mars-van Krevelen mechanism, followed by a rapid lattice oxygen replenishment to complete the reaction cycle. Our findings demonstrate that photocatalytic ODHE is a promising method for alkane-to-alkene conversions under mild conditions.

Ethylene is one of the most widely used feedstocks in the today's chemical industry. Currently, high-temperature steam cracking of large hydrocarbons derived from naphtha is the primary method for producing ethylene. The recent discovery of large reserves of shale gas, which contains about 10% ethane ($C_2H_6$), has sparked interest in ethane dehydrogenation as an alternative method for producing ethylene[1]. Researchers are now actively seeking new catalytic materials and technologies capable of efficiently and selectively converting ethane to ethylene.

In recent years, significant progress has been made towards the discovery of efficient catalysts for ethane dehydrogenation, with zeolite-derived and vanadium-based catalysts receiving a lot of attention[2,3]. Given that the C–H bond energy in $C_2H_6$ is exceptionally high (415 kJ $mol^{-1}$), harsh reaction conditions are typically required to achieve the first hydrogen atom extraction from ethane (the rate-limiting step for direct ethane dehydrogenation)[4]. Compared with direct ethane dehydrogenation, the oxidative dehydrogenation of ethane (ODHE), which utilizes an oxidant (e.g. $O_2$, $CO_2$, or $N_2O$), is more thermodynamically favorable (for example, $C_2H_6 + 0.5O_2 \rightarrow C_2H_4 + H_2O$, $\Delta G_r = -136.2$ kJ $mol^{-1}$) and can deliver both good ethane conversions and acceptable ethylene selectivities[5]. However, thermochemical ODHE still requires high reaction

[1]Key Laboratory of Photochemical Conversion and Optoelectronic Materials, Technical Institute of Physics and Chemistry, Chinese Academy of Sciences, Beijing 100190, China. [2]Center of Materials Science and Optoelectronics Engineering, University of Chinese Academy of Sciences, Beijing 100049, China. [3]Functional Crystals Lab, Technical Institute of Physics and Chemistry, Chinese Academy of Sciences, Beijing 100190, China. [4]School of Chemical Sciences, The University of Auckland, Auckland 1142, New Zealand. [5]Department of Chemical Engineering, University College London, London WC1E 7JE, UK. [6]Industrial Catalysis Center, Department of Chemical Engineering, Tsinghua University, Beijing 100084, China. ✉e-mail: shirun@mail.ipc.ac.cn; tierui@mail.ipc.ac.cn

temperatures ( > 500 °C) to achieve a meaningful ethane conversion, stimulating research towards alternative low-temperature ODHE processes powered by sustainable energy sources[6].

Photocatalysis holds great potential for the solar-driven activation of small molecules such as $O_2$, $CO_2$, and $CH_4$ under mild conditions. Metal oxide-based semiconductor photocatalysts, such as ZnO and $TiO_2$, have the ability to generate activated lattice oxygen upon illumination[7,8], enabling activation of inert chemical bonds via lattice oxygen-mediated photocatalytic oxidative pathways. For examples, Au-$ZnO/TiO_2$[7], Au(Pt, Pd, Ag)/ZnO[9], and Ag/ZnO[10] photocatalysts show activity for the photocatalytic methane coupling, partial oxidation, and combustion, respectively. These catalysts all use photogenerated active oxygen species ($O^-$) derived from $TiO_2$ or ZnO lattice oxygen for C−H bond activation. Moreover, the introduction of metal (metal alloy) nanoparticles with appropriate metal–support interaction boosts interfacial charge transfer and surface active adsorbent conversion, thus speeding up the catalytic reaction kinetics[11]. Tang et al. reported that AuCu−ZnO photocatalyst achieved methane partial oxidation for methanol and formaldehyde production with efficient charge transfer enhanced by Au and Cu species[12]. Whilst photocatalytic C−H bond activation in methane conversion has received considerable attention, no research has yet been conducted on photocatalytic ODHE for ethylene production, motivating a detailed investigation.

Herein, we synthesized PdZn intermetallic nanoparticles supported on ZnO (PdZn-ZnO) as a photocatalyst for ODHE with $O_2$. Under 365 nm illumination, a flow photocatalytic system containing PdZn-ZnO delivered a $C_2H_4$ production rate of 46.4 mmol $g^{-1}$ $h^{-1}$ with 92.6% selectivity. This level of performance was vastly superior to that of photocatalysts containing Pd or other intermetallic nanoparticles on ZnO, whilst also outperforming most high-temperature thermochemical ODHE processes. Our comprehensive characterization studies demonstrate that a robust PdZn-ZnO interface effectively enhances the photogeneration of $O^-$ (from ZnO lattice oxygen), which in turn activates the C−H bond in ethane. Further, the PdZn-ZnO metal-support interaction allows fast electron transfer for efficient dioxygen reduction and ZnO lattice oxygen replenishment. The combination of these processes significantly reduced the apparent activation energy for ODHE to only 18.4 kJ $mol^{-1}$, with PdZn-ZnO delivering an ethane conversion of nearly 20% with about 87% selectivity in simulated shale gas and showing feasibilities in the selective oxidative dehydrogenation of propane and butane. Results identify photocatalysis as a promising strategy for the selective production of ethylene from ethane at low temperatures.

## Results and Discussion
### Synthesis and characterization of PdZn-ZnO photocatalysts

The X-ray diffraction (XRD) pattern of the Pd-doped $Zn_5(CO_3)_2(OH)_6$ precursor is shown in Supplementary Fig. 1. All peaks can be assigned to monoclinic hydrozincite, $Zn_5(CO_3)_2(OH)_6$ (JCPDS No. 19-1458)[13]. No peaks were seen for any Pd-containing species, indicating that Pd was highly dispersed in the sample[14]. Transmission electron microscopy (TEM) revealed Pd-doped $Zn_5(CO_3)_2(OH)_6$ possessed a nanosheet-like structure, with energy dispersive X-ray spectroscopy (EDS) confirming that Zn, O, and Pd were uniformly dispersed in the sample (Supplementary Fig. 2). After annealing at 300 °C in a $H_2$ flow (10 vol.% $H_2$ in Ar) for 4 h, the precursor was transformed into a PdZn-ZnO catalyst. As shown in the XRD pattern (Fig. 1a), the PdZn-ZnO catalyst consists of hexagonal wurtzite ZnO (JCPDS No. 36-1451) and tetragonal intermetallic PdZn nanoparticles (JCPDS No. 06-0620)[15,16] After dissolving the ZnO support with dilute hydrochloric acid, the isolated PdZn nanoparticles show a XRD pattern matched well with the predicted PdZn intermetallic nanoparticle (Supplementary Fig. 3). The Pd loading in PdZn-ZnO was determined to be 1.57 wt.%, which was close to the expected nominal value (2.0 wt.%) (Supplementary Table 1). The high-angle annular dark-field scanning TEM (HAADF-STEM) image and EDS

element maps confirmed the presence of PdZn intermetallic nanoparticles dispersed on ZnO nanoparticles (Fig. 1b, c). Lattice fringes with spacings of 0.28, 0.25, and 0.22 nm corresponded to ZnO (100), ZnO (101) and PdZn (101) planes, respectively[17,18]. For comparison, photocatalysts consisting of pristine ZnO and ZnO-supported Pd nanoparticles (Pd-ZnO) were prepared. The structure, morphology, and specific surface area of ZnO and Pd-ZnO photocatalysts were similar to that of PdZn-ZnO (Supplementary Fig. 4 to 6).

The (100) crystal plane of PdZn shows an interphase arrangement of Pd and Zn atomic columns and contains of (001) and (010) planes (Fig. 1d). The aberration-corrected HAADF-STEM image of a PdZn nanoparticle (Fig. 1e) in PdZn-ZnO shows highly ordered rectangular arrays with alternating bright and dark columns of atoms. The fast Fourier-transform (FFT) pattern is consistent with the simulation results for a PdZn intermetallic nanoparticle in the [010] direction (Supplementary Fig. 7). The corresponding intensity profile presents a periodic oscillation pattern in two perpendicular directions corresponding to [001] (Fig. 1f). The regular atomic configurations of Pd and Zn atoms in the PdZn nanoparticle were also evident in the corresponding elemental maps (Fig. 1g), further confirming the presence of PdZn intermetallic nanoparticles in PdZn-ZnO.

### Photocatalytic ODHE performance evaluation

Photocatalytic ODHE tests were carried out using a home-made continuous flow photoreactor equipped with a 365 nm LED lamp as the light source (see Methods for experimental details). A mixture of $C_2H_6$ (5 vol.% in Ar, 18 mL $min^{-1}$) and $O_2$ (1 vol.% in Ar, 12 mL $min^{-1}$) with a total flow rate of 30 mL $min^{-1}$ was introduced as the feed gas. Figure 2a and Supplementary Table 2 show the photocatalytic $C_2H_4$ production rate and $C_2H_4$ selectivity for PdZn-ZnO and a series of reference samples. PdZn-ZnO exhibited a $C_2H_4$ production rate of 46.4 mmol $g^{-1}$ $h^{-1}$ with an ethylene selectivity of 92.6%. No liquid carbon-containing products, such as methanol and ethanol, were formed during the photocatalytic reaction (Supplementary Fig. 8). The optimal photocatalyst dosage and nominal Pd loading were determined to be 5.0 mg and 2.0 wt.%, respectively (Supplementary Fig. 9 to 11). The calcination temperature of the precursor was optimized to be 300 °C (Supplementary Fig. 12 and 13). The Pd-ZnO reference delivered a $C_2H_4$ production rate of 15.6 mmol $g^{-1}$ $h^{-1}$ with 83.4% ethylene selectivity, which is significantly lower than PdZn-ZnO. It should be noted that for pristine ZnO, only $CO_2$ and traces of $C_2H_4$ were generated, with no obvious enhancement observed for a physical mixture of ZnO with PdZn nanoparticles (denoted as PdZn-ZnO-mix, Supplementary Fig. 14). We also prepared other intermetallic nanoparticles (AgZn, AuZn, PtZn, CuZn) supported on ZnO (denoted as MZn-ZnO, M = Ag, Au, Pt, and Cu) using methods similar to that used to prepare PdZn-ZnO (Supplementary Fig. 15). However, $CO_2$ was the major product for all these photocatalysts. In addition, catalysts containing PdZn nanoparticles on other metal oxide supports, including $TiO_2$, $In_2O_3$, $CeO_2$, and $Al_2O_3$ were prepared (Supplementary Fig. 16). PdZn-$TiO_2$ prefer to over-oxidize $C_2H_6$ to $CO_2$ with a low $C_2H_4$ selectivity of 47.6%. PdZn-$In_2O_3$, PdZn-$CeO_2$, and PdZn-$Al_2O_3$ showed negligible activity under the same reaction conditions. Clearly, the ZnO support actively participated in photocatalytic ODHE over PdZn-ZnO. Above results indicated that PdZn intermetallic nanoparticles, ZnO, and the interfacial contact between PdZn and ZnO were indispensable to the outstanding photocatalytic performance.

Figure 2b shows that the $C_2H_6/O_2$ ratio had a strong influence on the activity and selectivity of PdZn-ZnO for ethane-to-ethylene conversion. The $C_2H_4$ selectivity decreased from nearly 100% to 67.7% as the $C_2H_6$ (5 vol.% in Ar)/$O_2$ (1 vol.% in Ar) feed gas ratio was changed from 30/0 to 6/24 (total flow rate = 30 mL $min^{-1}$). The production rate of $C_2H_4$ followed a volcano-type relationship, with the highest rate achieved at a feed gas ratio of 18/12. The effect of the total flow rate was then investigated at the $C_2H_6/O_2$ ratio of 18/12 (Supplementary Fig. 17). The $C_2H_4$ production rate increased gradually from 5 mL $min^{-1}$ to

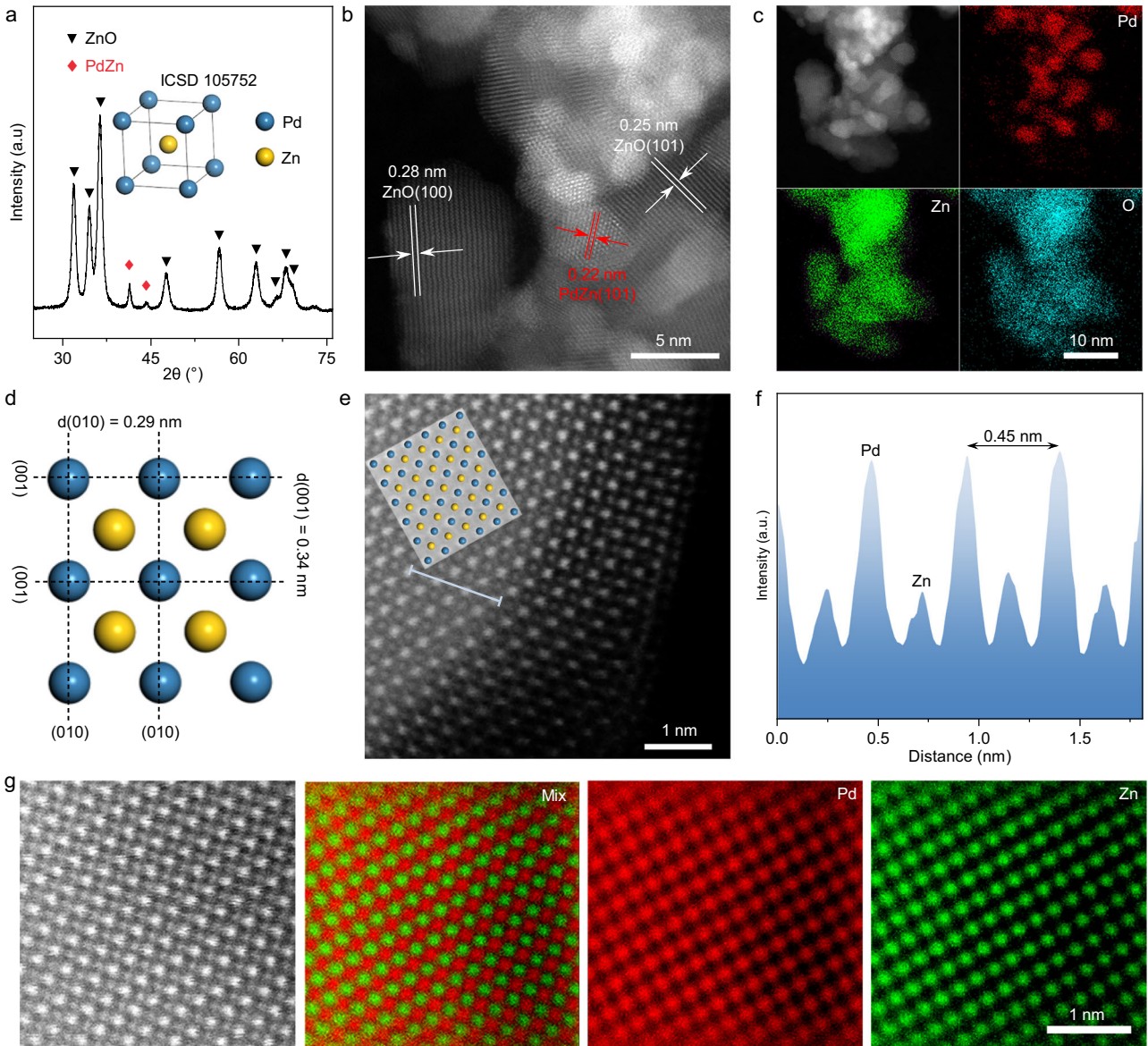

**Fig. 1 | Structural characterization of PdZn-ZnO. a** XRD patterns. Inset: crystal structure of PdZn. **b** HAADF-STEM image. **c** Energy-dispersive X-ray element maps. **d** Projection in the [010] direction for tetragonal PdZn. The yellow and blue spheres represent Zn and Pd atoms, respectively. **e** Aberration-corrected HAADF-STEM image. Inset: schematic of the PdZn intermetallic structure. **f** Intensity profiles measured from (**e**). **g** Aberration-corrected HAADF-STEM image and corresponding atomic-resolution elemental maps.

30 mL min$^{-1}$, reaching a maximum of 46.4 mmol g$^{-1}$ h$^{-1}$. The selectivity to C$_2$H$_4$ remained above 92% under these flow conditions. For a batch reactor (flow rate = 0), the best performance was realized after about 180 s of irradiation (15.0% C$_2$H$_6$ conversion, 6579.3 µmol g$^{-1}$ C$_2$H$_4$ production, and 85% ethylene selectivity, Supplementary Fig. 18), with the short time to reach the optimum being due to depletion of O$_2$ in the batch reactor. At higher reactant concentrations, the ethylene production rate exhibited a substantial increase, reaching up to 225.9 mmol g$^{-1}$ h$^{-1}$. Notably, the ethylene selectivity showed a gradual decline from 92.6% to 61.9% as the concentration of reactants increased (Supplementary Fig. 19). This suggests the importance of finding a balance between the rate of product formation and selectivity. During a 12 h of continuous photocatalytic stability test under the optimized reaction condition, the production rate of C$_2$H$_4$ over PdZn-ZnO fluctuated around 45 mmol g$^{-1}$ h$^{-1}$ with no obvious deactivation and a stable ethylene selectivity of around 92% (Fig. 2c). The spent catalyst showed almost no structural changes as confirmed by XRD (Supplementary Fig. 20). The nearly 100% carbon balance, supported by thermogravimetric and Raman spectroscopy analyses, confirms the absence of carbon deposition (Supplementary Fig. 21-23).

As shown in Fig. 2d and Supplementary Table 3, the photocatalytic ODHE performance of PdZn-ZnO was superior to representative thermocatalysts reported to date when considering the three main reaction parameters: temperature, C$_2$H$_4$ production rate, and selectivity[19–30]. Moreover, the as-developed PdZn-ZnO photocatalytic system also demonstrated exceptional performance for the oxidative dehydrogenation of propane (a propene production rate of 54.0 mmol g$^{-1}$ h$^{-1}$ with 89.4% propene selectivity) and butane (a butene production rate of 59.1 mmol g$^{-1}$ h$^{-1}$ with 95.9% butene selectivity) (Supplementary Fig. 24). Additionally, by replacing the ethane feed gas with a simulated shale gas (45 vol.% CH$_4$, 5 vol.% C$_2$H$_6$, balanced with Ar), a C$_2$H$_4$ production rate of about 60 mmol g$^{-1}$ h$^{-1}$ was achieved, with about 87% ethylene selectivity at nearly 20% ethane conversion (Supplementary Fig. 25).

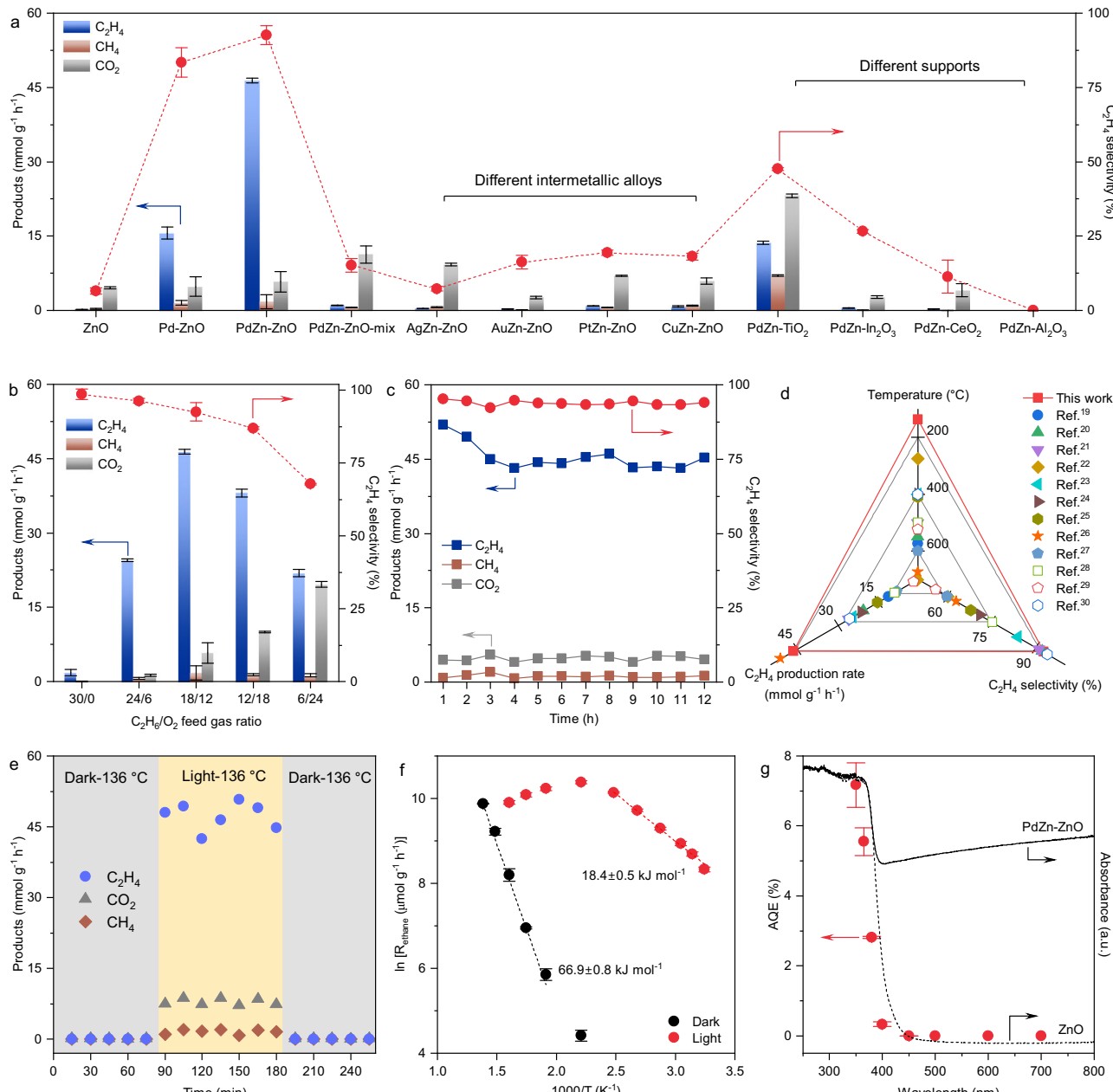

**Fig. 2 | Photocatalytic ODHE performance of PdZn-ZnO. a** ODHE tests over PdZn-ZnO and comparison photocatalysts. Reaction condition: 5.0 mg photocatalyst, $C_2H_6$ (5 vol.% in Ar, 18 mL min$^{-1}$) + $O_2$ (1 vol.% in Ar, 12 mL min$^{-1}$), 365 nm LED, 600 mW cm$^{-2}$. **b** Photocatalytic activity at a total flow rate of 30 mL min$^{-1}$ with different $C_2H_6/O_2$ feed gas ratios. **c** Photocatalytic stability test for 12 h of continuous irradiation. **d** Performance comparison with reported thermocatalysts.

**e** Photocatalytic activity at 136 °C under dark-light switch conditions. **f** Arrhenius plots measured in the dark and light conditions. Light intensity = 55.9 mW cm$^{-2}$. **g** Wavelength-dependent AQE and diffuse reflectance spectra of ZnO and PdZn-ZnO. Error bars represent standard deviations obtained from three independent measurements.

It is noteworthy that the surface temperature of the PdZn-ZnO photocatalyst increased under light irradiation due to the photo-thermal effects, and reached 136 °C at 600 mW cm$^{-2}$. (Supplementary Fig. 26a). As depicted in Fig. 2e, no significant product formation was observed by electrically heating the reactor containing PdZn-ZnO to 136 °C in the dark. However, on exposing the photocatalyst to light irradiation, the $C_2H_4$ production rate was greatly enhanced at the same temperature, and showed a linear relationship with the light intensity (Supplementary Fig. 26b). The results indicate that the ethane oxidative dehydrogenation process involved a photocatalytic mechanism. Arrhenius plots based on ethane reaction rates ($R_{ethane}$) at different temperatures are shown in Fig. 2g. The apparent activation energy for ODHE over PdZn-ZnO under light conditions (18.4 kJ mol$^{-1}$) was

significantly lower than in the dark (66.9 kJ mol$^{-1}$), implying that photons changed the reaction pathway for ODHE and significantly reduced the reaction energy barrier. $R_{ethane}$ values decreased under light conditions at temperatures greater than 180 °C, which is explained by shortened lifetime of photogenerated charge carriers and increased oxygen consumption due to ethane overoxidation at such high temperatures (Supplementary Fig. 27 and 28)[31]. The wavelength-dependent apparent quantum efficiency (AQE) for photocatalytic ODHE over PdZn-ZnO was found to be 7.2%, 5.5%, 2.8%, and 0.3% under monochromatic irradiation at 350, 365, 380, and 400 nm, respectively (Fig. 2f). This trend closely matched the absorbance spectrum of ZnO, indicating that photo-excitation of ZnO was a key process in ethylene production.

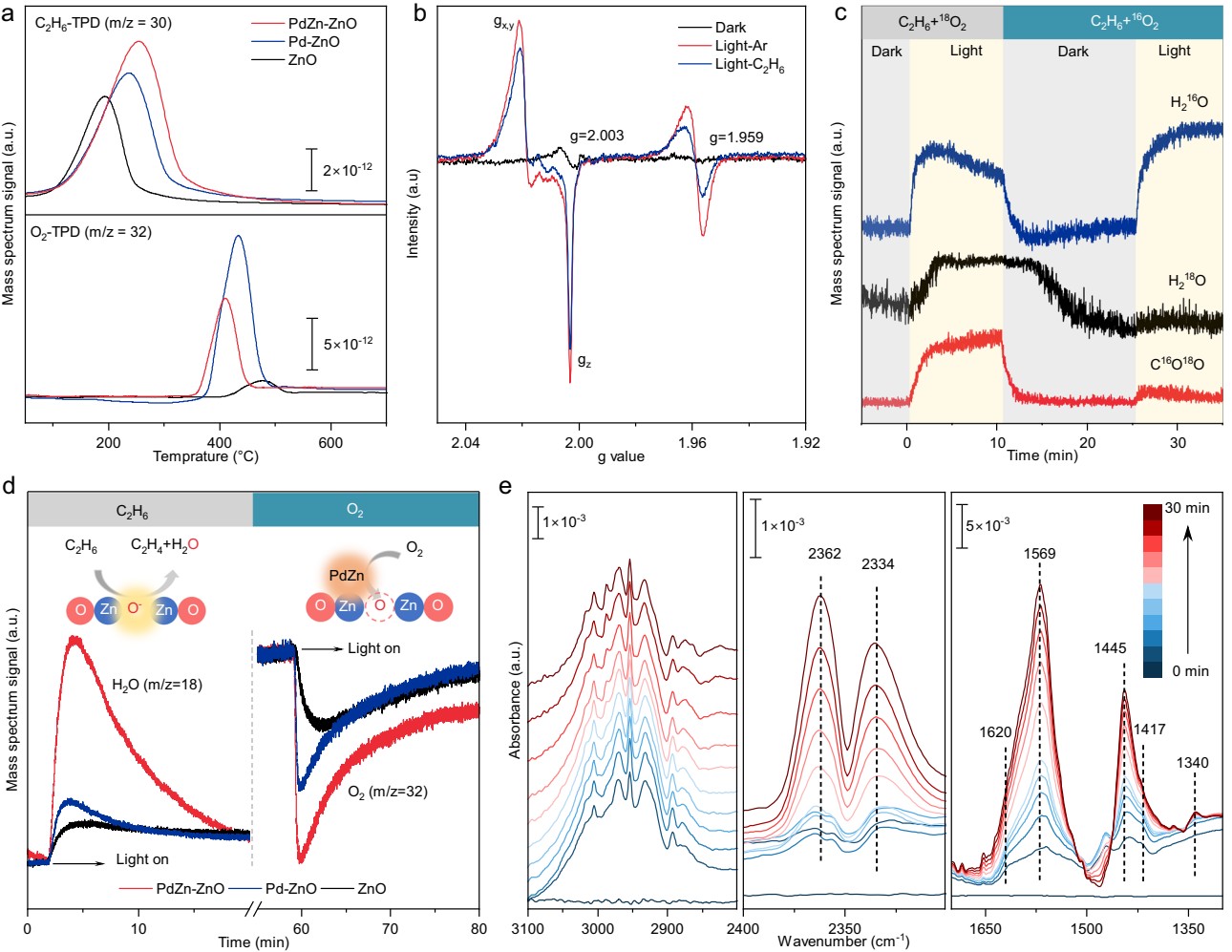

**Fig. 3 | Spectral studies for reaction intermediates of photocatalytic ODHE. a** $C_2H_6$-TPD and $O_2$-TPD spectra for ZnO, Pd-ZnO and PdZn-ZnO. **b** In situ EPR spectra for ZnO exposed to Ar in the dark and under 365 nm irradiation, and exposed to $C_2H_6$ under 365 nm irradiation. **c** Evolution of reactants and products during ODHE monitored by online MS. **d** Time-resolved online MS data for $C_2H_6$ dehydrogenation over ZnO, Pd-ZnO and PdZn-ZnO. **e** In situ FT-IR spectra collected from PdZn-ZnO in a feed gas (3 vol.% $C_2H_6$ + 0.4 vol.% $O_2$, balanced with Ar) at different irradiation times.

## Mechanistic investigations

PdZn-ZnO and two reference samples, ZnO and Pd-ZnO, were selected for further mechanistic investigations. Firstly, temperature-programmed desorption experiments for $C_2H_6$ and $O_2$ ($C_2H_6$-TPD and $O_2$-TPD) were conducted, with the results shown in Fig. 3a. The data show that $C_2H_6$ (m/z = 30) desorbed at 190.7, 233.8 and 254.6 °C on ZnO, Pd-ZnO, and PdZn-ZnO, respectively. Clearly, PdZn-ZnO shows a higher desorption temperature than the other samples, indicating a strong chemical adsorption towards $C_2H_6$. Furthermore, PdZn-ZnO demonstrates the highest ethane adsorption strength when compared to PdZn-TiO$_2$, PdZn-In$_2$O$_3$, PdZn-CeO$_2$, and PdZn-Al$_2$O$_3$ (Supplementary Fig. 29). This heightened ethane adsorption capability could be a key factor contributing to the significantly improved photocatalytic ODHE performance observed in PdZn-ZnO compared to its counterparts. In the $O_2$-TPD profiles, $O_2$ (m/z = 32) desorption signals were observed between 400 and 500 °C for all three samples, which can be ascribed to the desorption of chemisorbed oxygen species. The $O_2$ adsorption capacity of pristine ZnO was modest, and largely attributable to $O_2$ adsorption/dissociation on oxygen defects produced by sample pretreatment in a hydrogen atmosphere[32]. Once Pd or PdZn nanoparticles were introduced, the amount of adsorbed oxygen increased significantly, indicating that Pd and PdZn were the dominant active sites for $O_2$ adsorption and activation[33].

The efficient separation and transport of photogenerated charge carriers are critical for photocatalytic reactions[34]. The photoluminescence (PL) intensity under 365 nm excitation was notably decreased for Pd-ZnO and PdZn-ZnO compared with pristine ZnO, indicating improved separation of photo-excited electron-hole pairs (Supplementary Fig. 30). To investigate the carrier dynamics further, time-resolved transient absorption (TA) spectra were then recorded under 365 nm excitation and probed at 520 nm (Supplementary Fig. 31). The results showed that PdZn-ZnO had the longest lifetime of photogenerated charge carriers (187.3 ps), compared to Pd-ZnO (132.9 ps) and ZnO (118.6 ps), confirming its superior carrier separation efficiency. Next, in situ electron paramagnetic resonance (EPR) experiments were conducted to investigate the electron transfer and ethane activation processes. As shown in Fig. 3b and Supplementary Fig. 32, under dark conditions exposed to Ar atmosphere, both ZnO and PdZn-ZnO showed a symmetric EPR peak with a g value of 2.003, which could be attributed to unpaired electrons trapped at oxygen vacancies ($O_v$) in ZnO[35]. Under light irradiation, a symmetric EPR peak at g = 1.959 appeared for ZnO due to trapped photogenerated electrons (i.e. Zn$^+$ species)[36]. However, this trapped electron signal was weak for PdZn-ZnO, suggesting that the excited electrons in ZnO were transferred to PdZn nanoparticles, which acted as electron acceptors[37]. Additionally, anisotropic EPR signals, namely $g_x = 2.023$, $g_y = 2.019$, and

$g_z = 2.003$, also appeared under irradiation, which can be assigned to $O^-$ species created by the ZnO lattice oxygen (formally $O^{2-}$) trapping photogenerated holes[38]. After switching the contacting atmosphere from Ar to $C_2H_6$ under light conditions, the intensity of the EPR signal related to $O^-$ species decreased significantly for both ZnO and PdZn-ZnO, inferring the reaction of surface $O^-$ with $C_2H_6$ molecules. The results provide strong evidence for enhanced interfacial electron transfer from ZnO to PdZn in PdZn-ZnO, whilst also establishing the crucial role of photogenerated surface $O^-$ derived from lattice oxygen in $C_2H_6$ activation.

Oxygen isotope labeling tracing experiments were utilized to explore the key role of lattice oxygen in photocatalytic ODHE. Figure 3c shows the results of the $^{18}O$ isotope experiments, with online mass spectrometry (MS) used to detect $^{18}O$-containing products of ODHE (see Methods for more details). Firstly, a gas mixture containing $C_2H_6$ and $^{18}O_2$ was fed to the reactor in the dark to remove residual gas impurities in the flow chamber. Upon irradiation of PdZn-ZnO, strong signals due to $H_2^{16}O$ and $C^{16}O^{18}O$ were observed confirming that surface lattice oxygen ($^{16}O$) on ZnO was an active site for water production and ethane conversion during photocatalytic ODHE. This result was further corroborated by the $^{18}O$-tracer experiments (Supplementary Fig. 33), in which $^{18}O$ had been pre-incorporated in the ZnO lattice of PdZn-ZnO. After 10 min of illumination, the light was turned-off and the atmosphere switched to a gas mixture of $C_2H_6$ and $^{16}O_2$. The detection of $^{18}O$ in the oxygen-containing gas products ($H_2^{18}O$, $C^{16}O^{18}O$) confirmed that $^{18}O$ from the $^{18}O_2$ feed replenished $O_v$ during the first irradiation period and could participate in the following ODHE reaction. This lattice oxygen replenishment process was also confirmed by the light-induced oxygen isotope exchange effect between $^{18}O_2$ and PdZn-ZnO (Supplementary Fig. 34).

Time-resolved online MS spectra for $C_2H_6$ dehydrogenation over PdZn-ZnO, Pd-ZnO, and ZnO were then collected (Fig. 3d). An ethane gas flow was first introduced into the reactor, with a $H_2O$ signal being detected for all photocatalysts under light irradiation as a result of lattice oxygen consumption and $O_v$ generation on ZnO during the ODHE reaction. The light was then turned off and the atmosphere switched to $O_2$ for a period of time. When the light was again turned on, the signal for $O_2$ in the gas phase decayed, indicating the consumption of $O_2$ to fill $O_v$ sites. PdZn-ZnO showed the highest water generation and dioxygen consumption signals under illumination, indicating a significantly promoted lattice oxygen-mediated photocatalytic ODHE through a photo-enhanced Mars-van Krevelen (M-K) pathway. That is, the photogenerated surface $O^-$ from lattice oxygen on ZnO extracts hydrogen atoms from ethane, resulting in the formation of water molecules and $O_v$. Then, $O_2$ adsorbed on PdZn is activated by photo-excited electrons, with the $O_2$ molecule dissociating to fill $O_v$ and complete the reaction cycle.

In situ Fourier transform infrared (FT-IR) spectra were collected to gain additional information about the reaction intermediates and reaction mechanism (see Methods for experiment details). As shown in Fig. 3e, a series of absorption peaks appeared over PdZn-ZnO in a feed gas (3 vol.% $C_2H_6$ + 0.4 vol.% $O_2$, balanced with Ar) under light conditions, with the intensities of the peaks gradually increasing with irradiation time. A Zn–OH bending vibration at 1417 $cm^{-1}$ was detected[39], indicating the presence of surface-adsorbed hydroxyl groups (*OH) generated by hydrogen atom extraction from ethane on surface lattice oxygen (i.e. photo-generated $O^-$ sites) of ZnO[40]. The absorption peaks at 1445/1620 $cm^{-1}$, together with the absorption band around 2950 $cm^{-1}$, could be assigned to C = C and C–H stretching vibrations of adsorbed $C_2$ intermediates (*$C_2H_{4+n}$, n = 0, 1, 2)[41,42]. The peaks at 1569 $cm^{-1}$ and 1340 $cm^{-1}$ were assigned to adsorbed carboxyl species (HCOO*) species associated with ethane overoxidation products, which were also evidences by gaseous $CO_2$ peaks at 2334 $cm^{-1}$ and 2362 $cm^{-1}$ [43]. The FT-IR results demonstrate the formation of surface-adsorbed *OH, *$C_2H_{4+n}$, and HCOO* during ODHE over PdZn-ZnO.

These are the expected intermediates for lattice oxygen-mediated photocatalytic ODHE (along with some minor intermediates and products associated with the over-oxidation of ethane).

To gain a better understanding of the electron transfer processes and reaction transition states (TS) occurring at the interface between the metal nanoparticle (PdZn or Pd) and the ZnO support, density functional theory (DFT) calculations were performed. Two models were built and optimized to simulate the structure of PdZn-ZnO and Pd-ZnO. A $Pd_{15}Zn_{15}$ cluster supported on ZnO(101) (denoted as $Pd_{15}Zn_{15}$-ZnO) was used to represent PdZn-ZnO, whilst a $Pd_{30}$ cluster supported on ZnO(101) (denoted as $Pd_{30}$-ZnO) was used to represent Pd-ZnO, as shown in Fig. 4a, d. A charge density difference comparison between the two models revealed that charge redistribution is localized in a limited region near the interface between metal nanoparticles and ZnO supports (Fig. 4b, e). Bader charge analysis predicted increased charge accumulation ($+8.32\ q_e$) on the $Pd_{15}Zn_{15}$ cluster (Fig. 4c) compared to the $Pd_{30}$ cluster ($+5.82\ q_e$, Fig. 4f), indicating that electron transfer from PdZn to ZnO was enhanced compared to electron transfer from Pd to ZnO. The higher binding energy of the Pd 3d signal in PdZn-ZnO than in Pd-ZnO also signifies a more pronounced electron deficiency state associated with interfacial electron transfer (Supplementary Fig. 35). DFT calculations also showed that the strong interfacial electron delocalization and transfer in $Pd_{15}Zn_{15}$-ZnO resulted in a more negative Pd d-band center position (−1.68 eV) compared with $Pd_{30}$-ZnO (−1.54 eV) (Fig. 4g). The more negative d-band center in the case of supported PdZn contributes to a downshifted antibonding state for oxygen adsorbates, leading to an increased d orbital occupancy and weakened adsorption energy of activated oxygen adsorbates[44,45]. This was expected to be beneficial for subsequent lattice oxygen replenishment processes on adjacent ZnO during ODHE.

The reaction kinetics of dioxygen activation and ethane dehydrogenation to ethylene over $Pd_{15}Zn_{15}$-ZnO and $Pd_{30}$-ZnO were then simulated, and the energy profiles are shown in Fig. 4h. Additional information on the models and energy values are summarized in Supplementary Fig. 36-37 and Supplementary Table 4-5. The adsorption energy of $O_2$ on $Pd_{15}Zn_{15}$-ZnO was found to be significantly higher (−2.35 eV) than that on $Pd_{30}$-ZnO (−1.54 eV). The adsorbed oxygen (*$O_2$) then dissociates to two oxygen atoms (*$O_2$ → 2*O, TS1), with activation barriers ($E_a$) of 0.68 eV and 1.08 eV on $Pd_{15}Zn_{15}$-ZnO and $Pd_{30}$-ZnO, respectively. These results are consistent with the rapid oxygen consumption over PdZn-ZnO observed on the time-resolved online MS (Fig. 3d).

$Pd_{15}Zn_{15}$-ZnO was found to have an $E_a$ of 0.32 eV for *O-assisted C–H bond scission in $C_2H_6$ (*O + *$C_2H_6$ → *$C_2H_5$ + *OH, TS2), which was 0.54 eV lower than that of $Pd_{30}$-ZnO, indicating that ethane activation was kinetically favored over PdZn-ZnO. Subsequently, the β-H of *$C_2H_5$ is extracted to form *$C_2H_4$ and *H on ZnO (TS3). Then, *OH and *H combine to form $H_2O$ (TS4) via two possible pathways, with the oxygen in $H_2O$ coming from either surface lattice oxygen (M-K pathway) or dissociated dioxygen (Langmuir-Hinshelwood, L-H pathway). The barriers of these two pathways over $Pd_{15}Zn_{15}$-ZnO were calculated to be 0.71 eV and 0.92 eV, respectively (Supplementary Fig. 38 and 39), suggesting that the M-K pathway is kinetically favored. The M-K pathway leaves an $O_v$ on ZnO and an additional *O on PdZn, respectively, followed by a ZnO lattice oxygen replenishment process between the two species (TS5) with an $E_a$ of 0.81 eV. $Pd_{30}$-ZnO exhibits larger $E_a$ for both TS4 and TS5 compared with $Pd_{15}Zn_{15}$-ZnO (Supplementary Fig. 40 and 41). We compared the energy profiles of the first two reaction steps (TS1 and TS2 as potential rate-determining steps) with models constructed for other Pd-based intermetallic nanoparticles on ZnO ($Pd_{15}In_{15}$-ZnO, $Pd_{15}Cu_{15}$-ZnO and $Pd_{15}Ni_{15}$-ZnO). Results showed that $Pd_{15}Zn_{15}$-ZnO exhibited the lowest $E_a$ for both oxygen activation and hydrogen extraction from ethane, validating its outstanding photocatalytic ODHE performance (Supplementary Fig. 42 and 43).

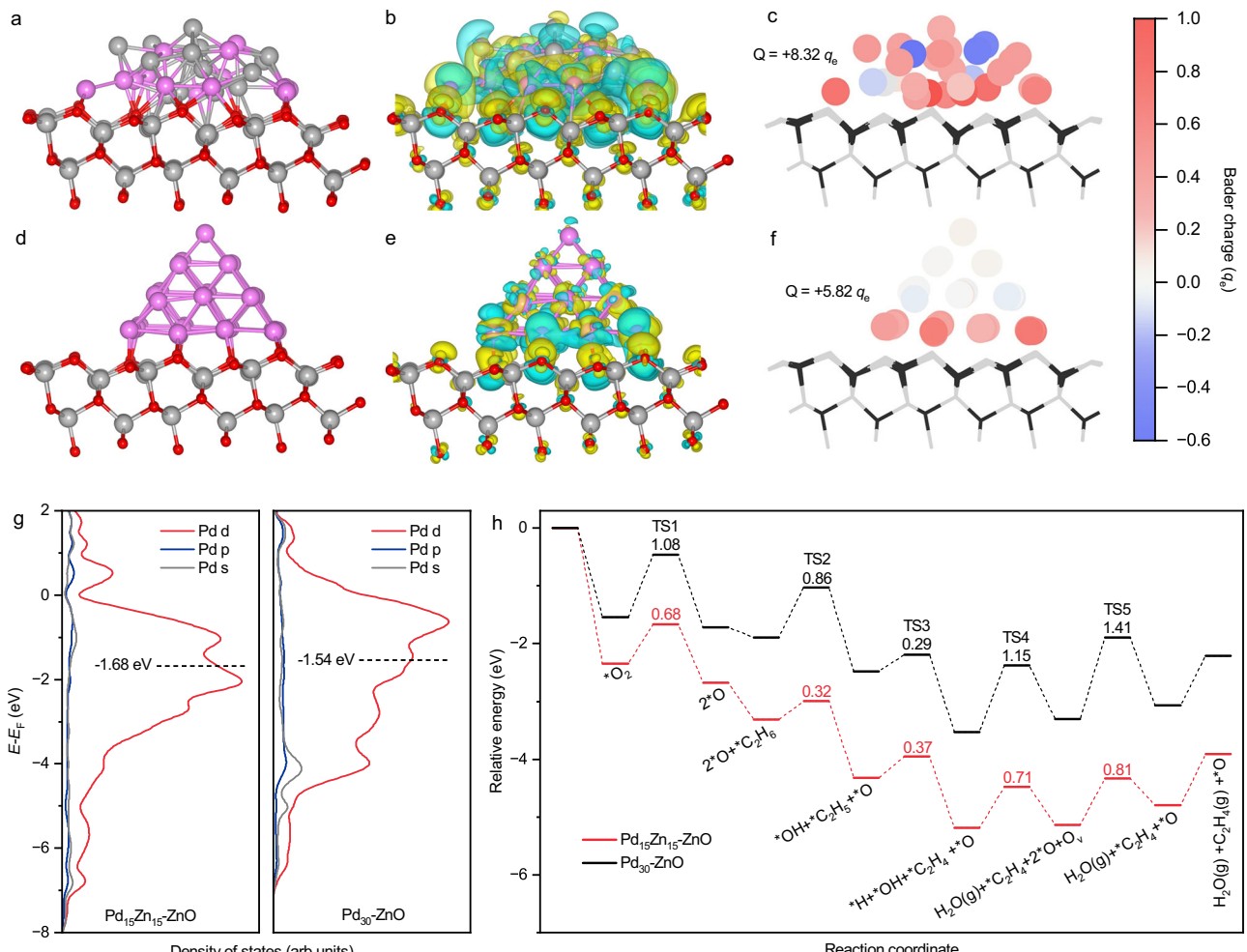

**Fig. 4 | DFT calculations exploring surface charge distributions and reaction kinetics. a** and **d** Model structures for $Pd_{15}Zn_{15}$-ZnO (**a**) and $Pd_{30}$-ZnO (**d**). **b** and **e** Charge density difference plots for $Pd_{15}Zn_{15}$-ZnO (**b**) and $Pd_{30}$-ZnO (**e**). The Pd, Zn, and O atoms are shown in purple, gray, and red. The yellow and cyan surfaces correspond to regions of charge gain (accumulation) and loss (depletion). The isovalue of the isosurfaces is $3.0 \times e\,Å^{-3}$. (**c** and **f**) Bader charge distributions for $Pd_{15}Zn_{15}$-ZnO (**c**) and $Pd_{30}$-ZnO (**f**). The red and blue spheres indicate the extent of charge depletion (positive) and accumulation (negative). **g** Density of states of Pd atoms in $Pd_{15}Zn_{15}$-ZnO and $Pd_{30}$-ZnO. The horizontal dashed lines indicate the calculated d-band center. $E$-$E_F$ represents the energy relative to the Fermi energy level. **h** Calculated potential energy diagrams for ODHE to $C_2H_4$ on $Pd_{15}Zn_{15}$-ZnO and $Pd_{30}$-ZnO.

Based on the results presented, we propose a probable reaction mechanism for the selective photocatalytic ODHE over PdZn-ZnO (Fig. 5). Initially, photoexcited electron-hole pairs are generated in ZnO under UV (365 nm) irradiation. The photogenerated electrons in ZnO are efficiently transferred to PdZn nanoparticles, activating $O_2$ to adsorbed oxygen atoms. Meanwhile, the photogenerated holes reaching the surface lattice oxygen of ZnO to form $O^-$, which activates C−H bond scission in $C_2H_6$ to form $*C_2H_5$ and $*OH$. Subsequently, β-H cleavage in $*C_2H_5$ leads to the formation of $*C_2H_4$ and a water molecule, both of which desorb from the photocatalysts (leaving an $O_v$ on the ZnO surface). Finally, the $O_v$ is filled by the adsorbed oxygen atom spilling over from the PdZn sites.

In summary, we report the first example of selective photocatalytic ODHE with $O_2$ for ethylene production. A PdZn-ZnO photocatalyst, containing PdZn intermetallic nanoparticles, was prepared via thermal hydrogen reduction of a Pd-doped $Zn_5(CO_3)_2(OH)_6$ precursor. The strong electron transfer interactions at the PdZn-ZnO interface promotes the photogeneration of surface $O^-$ active sites on ZnO to activate the C−H bonds in ethane, leading to the formation of ethylene and water. Photo-excited electron transfer from ZnO to PdZn facilitated dioxygen dissociation, allowing fast replenishment of ZnO lattice oxygen consumed in the Mars-van Krevelen mechanism.

Photocatalytic ODHE on PdZn-ZnO had an apparent activation energy of $18.4\ kJ\,mol^{-1}$ under irradiation conditions, much lower than that in the dark ($66.9\ kJ\,mol^{-1}$). The PdZn-ZnO photocatalyst shows an ethylene production rate of $46.4\ mmol\,g^{-1}\,h^{-1}$ with 92.6% selectivity at low temperatures, exhibiting high alkene production rates and selectivity for photocatalytic dehydrogenation of propane and butane, as well as ethane in simulated shale gas. This work offers a proof-of-concept investigation, demonstrating that low-temperature photocatalytic ODHE is a promising new approach for ethylene production from ethane-containing feedstocks. Future work should be directed towards more detailed investigation of reaction mechanisms and also reactor designs for process scale-up.

## Methods

### Chemicals

$Zn(NO_3)_2\cdot6H_2O$, $Na_2CO_3$ and $Cu(NO_3)_2\cdot3H_2O$ were obtained from Beijing Innochem Science & Technology Co., Ltd. $Na_2PdCl_4$, $H_2PtCl_6$, $AgNO_3$, $HAuCl_4$ were purchased from Shanghai Aladdin Biochemical Technology Co., Ltd. $C_2H_6$ (5 vol.% in Ar), $O_2$ (1 vol.% in Ar) and a mixed reactant gas (3 vol.% $C_2H_6$, 1 vol.% $O_2$, balanced with Ar) were purchased from Beijing Beiyang United Gas Co., Ltd. $^{18}O$ isotopically labeled oxygen (1 vol.% $^{18}O_2$ in Ar) was obtained from Shanghai Maotoo

**Fig. 5 | Proposed reaction mechanism for photocatalytic ODHE over PdZn-ZnO.** Bond angles and bond lengths are not accurate in the schematic. The gray sphere represents a PdZn nanoparticle.

Specialty Gases Co., Ltd. All materials were used without further purification. Deionized water was used in the synthesis of all catalysts.

## Preparation of catalysts

(a) PdZn-ZnO. Pd-doped $Zn_5(CO_3)_2(OH)_6$ precursor (Pd-$Zn_5(CO_3)_2(OH)_6$) was synthesized according a commonly used coprecipitation method. Typically, for the preparation of the PdZn-ZnO photocatalyst, 2.97 g $Zn(NO_3)_2 \cdot 6H_2O$ and a certain amount of $Na_2PdCl_4$ were dissolved in deionized water. Then, a $Na_2CO_3$ aqueous solution was added dropwise to the metal salt solution under continuous stirring. Next, the Pd-$Zn_5(CO_3)_2(OH)_6$ product was collected by centrifugation, washed several times with deionized water and then finally vacuum freeze-dried for 24 h. Next, the Pd-$Zn_5(CO_3)_2(OH)_6$ precursor was heated at 5 °C min$^{-1}$ to 300 °C under a $H_2/Ar$ (10/90) flow (300 mL min$^{-1}$), then held at 300 °C for 4 h.

(b) MZn-ZnO. The synthesis methods for AuZn-ZnO, AgZn-ZnO, PtZn-ZnO, CuZn-ZnO photocatalysts (with the mass percentage of Au, Ag, Pt or Cu 2 wt.% relative to ZnO), were similar to those described for the PdZn-ZnO photocatalyst with different metals salts being used as required.

(c) ZnO and Pd-ZnO. Pristine ZnO photocatalyst was prepared by heating $Zn_5(CO_3)_2(OH)_6$ at 300 °C in a $H_2/Ar$ (10/90) flow (300 mL min$^{-1}$) for 4 h. The Pd-ZnO photocatalysts was synthesized using a $NaBH_4$ reduction method[46] with the above synthesized ZnO as raw material, followed by heating in a nitrogen gas stream at 300 °C for 4 h.

(d) PdZn-ZnO-mix. For the synthesis of PdZn-ZnO-mix sample, 1.0 g of PdZn-ZnO was immersed in 30 mL of an aqueous 1 mol L$^{-1}$ HCl solution for 10 min to dissolve the ZnO support, then collected by centrifugation and washed with water. The obtained PdZn nanoparticles was dispersed in 100 mL of deionized water together with 980 mg of ZnO. The PdZn and ZnO dispersion was sonicated for 30 min, followed by freeze-dried for 24 h.

## Characterizations

The structure and crystallinity of the photocatalysts were examined by X-ray diffraction (XRD, Bruker AXSD8 Advance, Germany). The diffractometer was equipped with a Cu Kα radiation source ($\lambda$ = 1.5405 Å) operating at 40 kV. Morphologies and structure of the photocatalysts were studied using TEM (JEM, 2100 F, Japan). The aberration-corrected HAADF-STEM images and corresponding high-resolution EDS analyses were performed using a JEOL JEM-ARM300F atomic resolution electron microscope with a double spherical aberration corrector.

Nitrogen adsorption/desorption isotherms were collected at 77 K on a Quadrasorb SI MP apparatus. Specific surface areas were calculated via the BET method. Diffuse reflectance spectra were recorded on a Cary 7000 (Agilent) spectrometer equipped with an integrating sphere attachment. The actual metal contents in the photocatalysts were determined by inductively coupled plasma-optical emission spectroscopy (ICP-OES, Varian 710).

$C_2H_6$-TPD profiles were obtained using an AutoChem II 2920 (Micromeritics Instrument Corporation). Photocatalysts (200.0 mg) were pretreated under a He atmosphere for 60 min at 150 °C, using a heating rate of 10 °C min$^{-1}$. After cooling to 50 °C, photocatalysts were kept under a pure $C_2H_6$ atmosphere for 60 min to achieve adsorption saturation. Next, the samples were kept under a He flow (50 mL min$^{-1}$) for 60 min to remove any weakly physically adsorbed ethane. Finally, the $C_2H_6$-TPD tests were carried out by heating the photocatalysts from 50 °C to 800 °C at a heating rate of 10 °C min$^{-1}$ under a He carrier gas, with a mass spectrometer used for $C_2H_6$ detection. The test method for $O_2$-TPD experiments was similar to $C_2H_6$-TPD tests, except that $C_2H_6$ is replaced by $O_2$.

In situ EPR spectra were collected on a Bruker E500 spectrometer. Before the EPR tests, the catalyst was pre-treated at 150 °C for 2 h under an Ar flow to remove any surface adsorbed species. 50.0 mg of photocatalyst was loaded into a quartz tube and evacuated. Then, Ar or $C_2H_6$ gas was introduced into the EPR tube. EPR spectra were recorded in the dark and light conditions at 110 K.

Time-resolved online MS experiments were performed in a homemade fixed-bed stainless steel flow reactor (volume = 56.0 mL) with a quartz window on the top for irradiation of photocatalysts. The reactor was coupled to a mass spectrometer (SPIMS 2000, Hexin mass spectrum). Before each test, 20.0 mg of photocatalyst was uniformly dispersed on a glass fiber membrane and the modified membrane was then pretreated in an Ar gas flow at 200 °C for 2 h to remove any impurities from the membrane.

In situ Fourier transform infrared spectroscopy (FT-IR) data were collected on a Bruker Vertex 70 V FT-IR spectrometer equipped with a narrowband HgCdTe detector and a transmission reaction chamber (Harrick) connected to an evacuation line ($\sim 10^{-7}$ mbar). 5.0 mg of PdZn-ZnO photocatalyst was pressed into a self-supported pellet (7.0 mm in diameter) and placed in the transmission chamber. In a typical in situ FT-IR experiment, the photocatalyst pellet was first purged with Ar (100 mL min$^{-1}$) for 30 min to remove impurities adsorbed on the surface. Then, reactant gas (3 vol.% $C_2H_6$, 0.4 vol.% $O_2$, balanced with Ar, 30 mL min$^{-1}$) was introduced into the chamber. After 30 min of reactant adsorption, a background FT-IR spectrum was then recorded. Further FT-IR spectra were then collected every minute in

the reaction atmosphere under illumination from a 150 mW continuous diode laser (375 nm), which served as the light source. Each spectrum was recorded by averaging 200 scans collected at a scanning velocity of 40 kHz and a resolution of 4 $cm^{-1}$.

## Photocatalytic activity measurements

Photocatalytic ODHE tests were conducted in a custom-built fixed-bed stainless steel flow reactor (volume = 56.0 mL) with a quartz window on the top for light irradiation (Supplementary Fig. 44). Typically, photocatalyst (5.0 mg) was uniformly spread on a glass fiber membrane (Whatman, GE Healthcare Life Sciences, catalog number 1823-047) and then placed in the reactor (parallel to the quartz window). Then, a reaction gas comprising $C_2H_6$ (5 vol.% in Ar, 18 mL $min^{-1}$) and $O_2$ (1 vol.% in Ar, 12 mL $min^{-1}$) with a total flow rate of 30 mL $min^{-1}$ was introduced to the reactor. After purging for 30 min to remove the air in the reactor. a 365 nm LED lamp (100 W 365 nm LED, Beijing Perfect-Light Technology Co., Ltd., PLS-LED100C) was applied as light source to drive the reaction. Gas samples in the outlet were analyzed using a gas chromatograph (Shimadzu GC-2014, Shimadzu Co., Japan) equipped with three channels. The first channel analyzed hydrocarbons in an HP PLOT $Al_2O_3$ column with He as a carrier gas and a flame ionization detector (FID). The second channel analyzed $CO_2$, $N_2$, Ar, $O_2$, $CH_4$, and CO with a combination of micropacket Haysep Q, H-N, and Molsieve 13× columns using He as the carrier gas and a thermal conductivity detector (TCD). The third channel analyzes $H_2$ using a micropacket HayeSep Q and Molsieve 5 Å column with $N_2$ as a carrier gas and a TCD detector. For the batch-reactor test, a batch type reactor (volume = 56.0 mL) was used. After evacuation of the reaction system, reaction gas (3 vol.% $C_2H_6$, 0.4 vol.% $O_2$, balanced with Ar) was introduced into the reactor until the pressure reached 0.3 MPa, after which the photocatalyst was exposed to the 365 nm LED light source.

The $C_2H_4$ selectivity is calculated according to the following equation.

$$C_2H_4 \text{ selectivity}(\%) = \frac{n(C_2H_4)}{n(C_2H_4) + \frac{1}{2} \times [n(CH_4) + n(CO_2)]} \times 100\% \quad (1)$$

where $n(C_2H_4)$, $n(CH_4)$ and $n(CO_2)$ represent the moles of $C_2H_4$, $CH_4$ and $CO_2$ at the outlet, respectively. The conversion rate is calculated based on products according to the following equation.

$$C_2H_6 \text{ conversion}(\%) = \frac{n(C_2H_4) + \frac{1}{2} \times [n(CH_4) + n(CO_2)]}{n_0(C_2H_6)} \times 100\% \quad (2)$$

where $n_0(C_2H_6)$ represents the moles of $C_2H_6$ at the inlet. The carbon balance is calculated according to the following equation.

$$\text{Carbon balance}(\%) = \frac{n(C_2H_4) + \frac{1}{2} \times [n(CH_4) + n(CO_2)] + n_1(C_2H_6)}{n_0(C_2H_6)} \times 100\% \quad (3)$$

where $n_1(C_2H_6)$ represent the moles of $C_2H_6$ at the outlet, respectively. The AQE was calculated according to the following equation.

$$AQE = \frac{n(\text{electrons})}{n(\text{photons})} \times 100\% \quad (4)$$

where $n$(electrons) and $n$(photons) represent the number of reacted electrons and the number of incident photons, respectively. According to the chemical equation ($2C_2H_6 + O_2 \rightarrow 2C_2H_4 + 2H_2O$, $2C_2H_6 + 7O_2 \rightarrow 4CO_2 + 6H_2O$), $n$(electrons) = $2n(C_2H_4) + 7n(CO_2)$, where $n(C_2H_4)$ and $n(CO_2)$, represent the moles of produced $C_2H_4$ and $CO_2$, respectively. $n$(photons) = $IAt/E$, where $I$, $A$, t, and $E$ represent light intensity (W $cm^{-2}$), irradiation area (12.57 $cm^{-2}$), irradiation time (s) and photon energy (J), respectively. $E = hc/\lambda$, where $h$, $c$, and $\lambda$ represent

Planck's constant, light speed, and monochromatic light wavelength, respectively. The light intensity $I$ at different wavelengths ($\lambda = 350, 365, 380, 400, 450, 500, 600,$ and 700 nm) was measured to be 51.3, 52.6, 50.2, 49.8, 50.4, 50.5, 51.3, and 49.8 mW $cm^{-2}$ by xenon lamp (300 W Xe lamp, Beijing Perfectlight Technology Co. Ltd, PLS-SXE300D) source and band pass filter (Supplementary Fig. 45). Since the surface temperature of the catalyst was around room temperature under monochromatic light irradiation, the contribution of heat induced by illumination during AQE tests was not considered.

## Computational details

First-principles spin-polarized calculations were performed using the Vienna ab initio Simulation Program (VASP)[47,48]. The generalized gradient approximation (GGA) in the Perdew-Burke-Ernzerhof (PBE) form and a cutoff energy of 700 eV for planewave basis set were adopted[49]. A 3 × 3 × 1 Monkhorst-Pack[50] k grid was used for sampling the Brillouin zones in the structure calculations. Ion-electron interactions were described by the projector augmented wave (PAW) method[51]. The convergence criteria for structure optimization was a maximum force less than 0.02 eV $Å^{-1}$ on each atom with an energy change less than $1 \times 10^{-5}$ eV. The DFT-D3 semiempirical correction was described via Grimme's scheme method[52,53].

Since the main exposed facets of ZnO were (101), a ZnO(101) model was used for the calculations. The ZnO(101) surface was modelled by a 3 × 3 unit cell based on an initial structure with lattice parameters a = 18.4 Å, b = 19.5 Å, c = 25.4 Å, α = β = 90°, γ = 105°. $Pd_{15}Zn_{15}$ cluster with fifteen Pd atoms and fifteen Zn atom was used to simulate a PdZn intermetallic nanoparticle, whilst $Pd_{30}$ cluster with thirty Pd atoms was used to simulate a Pd nanoparticle. The above $Pd_{15}Zn_{15}$ and $Pd_{30}$ clusters were placed on ZnO(101) surface to form models $Pd_{15}Zn_{15}$-ZnO(101) and $Pd_{30}$-ZnO(101), respectively. Models similar to $Pd_{15}Zn_{15}$-ZnO(101) were used to build $Pd_{15}Cu_{15}$-ZnO(101), $Pd_{15}Ni_{15}$-ZnO(101), and $Pd_{15}In_{15}$-ZnO(101).

To calculate the kinetic energy barrier of chemical reactions, the climbing image nudged elastic band (CI-NEB) method was used to search for the TS[54,55], with convergence criteria a force below 0.05 eV $Å^{-1}$. $E_a$ was calculated using $E_a = E_{TS} - E_{IS}$, where $E_{TS}$ and $E_{IS}$ represents the energy of TS and the initial state.

## Data availability

The datasets generated and/or analyzed during the current study are available from the corresponding author on reasonable request. Received: ((will be filled in by the editorial staff)) Accepted: ((will be filled in by the editorial staff)) Published online: ((will be filled in by the editorial staff)) Source data are provided with this paper.

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

## Acknowledgements

The authors are grateful for financial support from the National Key R&D Program of China (2023YFA1507201), the National Natural Science Foundation of China (51825205, 52120105002, 22272190, 22088102), the Beijing Natural Science Foundation (2222035), the CAS Project for Young Scientists in Basic Research (YSBR-004), the DNL Cooperation Fund, CAS (DNL202016), and the Youth Innovation Promotion Association of the CAS. GINW acknowledges funding support from the Royal Society Te Apārangi (for a James Cook Research Fellowship), the MacDiarmid Institute for Advanced Materials and Nanotechnology, and the Energy Education Trust of New Zealand. The authors would like to thank Yueru Wu from Shiyanjia Lab (www.shiyanjia.com) for assistance with the TPD experiments.

## Author contributions

R.S. and T.Z. supervised the project. P.W. conceived the idea for the project and performed catalyst synthesis, characterization and photocatalytic tests. X.Z. participated in data processing for DFT calculations. J.Z. and R.S. assisted in data analysis. P.W. wrote the manuscript. R.S., G.I.N.W., J.T. and T.Z. revised the manuscript. All authors discussed the results and commented on the manuscript at all stages.

## Competing interests

The authors declare no competing interests.
