## [Peer Review File · Nature Communications]

Photocatalytic ethylene production by oxidative dehydrogenation of ethane with dioxygen on ZnO-supported PdZn intermetallic nanoparticlesREVIEWER COMMENTS

Reviewer #1 (Remarks to the Author):

This work presents the photocatalytic performance of PdZn intermetallic nanoparticles over ZnO. The authors carried out a lot of characterization, including isotope experiments and other in-situ experiments. The ethylene production rate of $46.4 \text{ mmol g}^{-1} \text{ h}^{-1}$ seems superior to others.

These kind reports usually focus on the material properties and reaction mechanism, while few reports on the reaction behaviors analysis. In this work, the author discussed the reaction behaviors. The overall structure and quality of the manuscript is satisfactory. However, the reaction behaviors and the reaction mechanism proposed in this work still need some modifications. Details comments are given below.

1. Why are the reactivities of different carriers different in the reaction in Fig. 2a ? What is the difference between the photoresponse performance and the basic adsorption performance of different carriers?
2. Since the reaction activity of the system is relatively low, the main concentration in the reactor is basically unchanged during the whole reaction. The photocatalytic behaviors of different raw material concentration should be further discussed, maybe compared with the thermocatalytic reactions.
3. Is there carbon deposit in the spend catalyst?
4. As shown in Fig. 2f, the activation energy regression of photocatalysis can be divided into two stages. Why is the high temperature stage not discussed?
The photoactivation energy of the reaction (18.4 KJ/mol) is obviously lower than that of the common chemical reaction. It is even close to the chemisorption energy. I think the external field energy is not calculated in this regression. Maybe the activation energy regressed here should be ascribed to mass transfer barriers.
5. Should the label of Fig. 3c(10-30 mins) be addressed as $\text{C}_2\text{H}_6 + 16\text{O}_2$?
6. In Fig.5, the energy of the dehydration process to form the O vacancy perhaps seems higher than the deoxidation from the PdZn alloy active site. Maybe the energy barriers should be compared form the calculation results. Or the configuration diagram of this two intermediates should be redrawn.

Reviewer #2 (Remarks to the Author):

In this manuscript, the authors reported a novel ethylene production pathway through photocatalytic oxidative dehydrogenation of ethane with oxygen under mild conditions, showing great development potential compared to the existing naphtha cracking route with high energy consumption and greenhouse gas emissions. In addition, the mechanism of photocatalytic oxidative dehydrogenation of ethane was studied, and a light-driven lattice oxygen-mediated activation mechanism of ethane was proposed. This work will provide a reliable guide for studying photocatalytic high-value-added conversion and activation mechanisms for alkanes. I think this manuscript can be accepted by Nature Communications after some minor revisions. Authors should carefully consider the following comments and suggestions.

- 1) In Fig. 2f, how does the selectivity of ethylene change with temperature? The decrease in the conversion rate of ethane at high temperatures may also be due to the excessive oxidation of ethylene at high temperatures and the consumption of oxygen.
- 2) The charge transfer behavior between PdZn nanoparticles and ZnO in PdZn-ZnO samples may be experimentally supported by X-ray photoelectron spectroscopy. I suggest that the author test the XPS profile of the sample to supplement it.
- 3) Why is the calcination temperature of the Pd-Zn₅(CO₃)₂(OH)₆ precursor selected at 300°C? Have the authors tried to treat the precursors at other temperatures?
- 4) Can the precursor decompose completely at 300 °C? I suggest that the author supplement the thermogravimetric analysis test of the sample.
- 5) Both photogenerated electrons and holes seem to play important roles in the photocatalytic oxidative dehydrogenation of ethane. It is suggested to eliminate photogenerated electrons or photogenerated holes with trapping agents such as AgNO₃ and CH₃OH to investigate their catalytic performance.

Reviewer #3 (Remarks to the Author):

Low-temperature oxidative dehydrogenation of C_2H_6 is an appealing approach to the synthesis of C_2H_4 but remains a challenge. In the manuscript, authors presented a PdZn-ZnO photocatalyst for C_2H_6 dehydrogenation to C_2H_4 by O_2 with high activity and selectivity, showing a remarkable superior to the currently-reported thermocatalytic system. Indeed, it is an interesting work. However, this work lacks more detailed experimental evidences and characterizations and theoretical calculations to reveal the reaction mechanism. Hence, authors should address the following points before publication.

The more detailed comments as follows:

1. Page 2, line 28-29, the optimized C_2H_6 conversion rate enables reach 20%, in which the conversion rate is calculated based on products or C_2H_6 before and after reaction? The carbon balance should be included in this work.
2. The specific role of Pd and Zn during the C_2H_6 dehydrogenation?
3. The enhanced activity is attributed to photocatalysis or photo-assisted thermocatalysis or synthetic effect of photo-thermal synergy? The specific role of photo and thermal? How the photogenerated electron and hole affect the reaction route?
4. The radical such as ($\cdot O_2^-$) should be detected during the reaction for excluding the radical mechanism.
5. The catalyst can drive the reaction under light at 136 °C. How about at room temperature? The specific role of thermal?
6. The MS signals of H_2 should be provided during the in-situ C_2H_6 dehydrogenation process detected by MS.
7. In Fig. 2c, compared with CH_4 and CO_2 , the C_2H_4 productivity fluctuates significantly while the C_2H_4 selectivity remains well. Please explain this phenomenon.

Response to the reviewers' comments

Reviewer #1: *This work presents the photocatalytic performance of PdZn intermetallic nanoparticles over ZnO. The authors carried out a lot of characterization, including isotope experiments and other in-situ experiments. The ethylene production rate of 46.4 mmol g⁻¹ h⁻¹ seems superior to others. These kind reports usually focus on the material properties and reaction mechanism, while few reports on the reaction behaviors analysis. In this work, the author discussed the reaction behaviors. The overall structure and quality of the manuscript is satisfactory. However, the reaction behaviors and the reaction mechanism proposed in this work still need some modifications. Details comments are given below.*

Response: Thank you for your professional comments. We have responded to the comments point by point and made some changes accordingly in the revised manuscript.

1) Why are the reactivities of different carriers different in the reaction in Fig. 2a? What is the difference between the photoresponse performance and the basic adsorption performance of different carriers?

Response: The diffuse reflectance spectra of the carriers indicate that ZnO, TiO₂, In₂O₃, and CeO₂ all possess pronounced light absorption properties at 365nm, whereas Al₂O₃ exhibits minimal absorption of ultraviolet and visible light (Fig. R1a). Temperature programmed desorption spectra of ethane reveal that ZnO-supported PdZn exhibits the highest desorption temperature for ethane (Supplementary Fig. 29 and Fig. R1b). This can be attributed to the distinctive adsorption and activation capabilities of ZnO for C–H bonds (*Nat. Catal.* 4, 2021, 1032–1042; *Energy Environ. Sci.* 2018, 11, 294–298). In summary, the variation in the photocatalytic ODHE (Oxidative Dehydrogenation of Ethane) performance among different carriers may stem from their diverse ethane adsorption capacities.

We add a corresponding discussion in line 192, “Furthermore, PdZn-ZnO demonstrates the highest ethane adsorption strength...”.

Fig. R1. (a) Diffuse reflectance spectra of ZnO, TiO₂, In₂O₃, CeO₂, and Al₂O₃. (b) Temperature programmed desorption spectra of ethane for PdZn-ZnO, PdZn-TiO₂, PdZn-In₂O₃, PdZn-CeO₂ and PdZn-Al₂O₃.

2) Since the reaction activity of the system is relatively low, the main concentration in the reactor is basically unchanged during the whole reaction. The photocatalytic behaviors of different raw material concentration should be further discussed, maybe compared with the thermocatalytic reactions.

Response: We evaluated the photocatalytic ODHE performance at different concentrations of reactants (ethane and oxygen). At the higher reactant concentrations, the ethylene production rate exhibited a substantial increase, reaching up to 225.9 mmol g⁻¹ h⁻¹. Notably, the ethylene selectivity showed a gradual decline from 92.6% to 61.9% with increasing reactant concentrations (Supplementary Fig. 19 and Fig. R2). This suggests the need to strike a balance between the rate of product formation and selectivity.

We add a corresponding discussion in line 148, “At higher reactant concentrations, the ethylene production rate...”.

Fig. R2. Photocatalytic ODHE performance at varied reactant concentrations.

3) Is there carbon deposit in the spent catalyst?

Response: Thermogravimetric analysis and Raman spectroscopy (Supplementary Fig. 21, 23, and Fig. R3) conducted on the spent catalyst revealed the absence of carbon deposition. Furthermore, the carbon balance, approximately 100%, also supports the conclusion that carbon deposition did not occur (Supplementary Fig. 22 and Fig. R4).

It is noteworthy that carbon deposition is an endothermic reaction, carbon deposition can be largely suppressed at reaction temperatures as low as 136 °C, which is one of the advantages of photocatalytic ODHE under mild conditions.

We add a corresponding discussion in line 156, “The nearly 100% carbon balance...”

Fig. R3. Raman spectra of fresh and spent PdZn-ZnO.

Fig. R4. Carbon balance for stability test in Fig. 2c.

4) As shown in Fig. 2f, the activation energy regression of photocatalysis can be divided into two stages. Why is the high temperature stage not discussed?

The photoactivation energy of the reaction (18.4KJ/mol) is obviously lower than that of the common chemical reaction. It is even close to the chemisorption energy. I think the external field energy is not calculated in this regression. Maybe the activation energy regressed here should be ascribed to mass transfer barriers.

Response: At elevated temperatures under light irradiation, the ethylene selectivity undergoes a notable reduction, signifying the gradual predominance of ethane overoxidation to carbon dioxide at this stage (Supplementary Fig. 28 and Fig. R5). Consequently, our primary focus is on the activation energy of the oxidative dehydrogenation of ethane to ethylene at lower temperatures.

In complex reactions such as the oxidative dehydrogenation of alkane, researchers often gauge the catalyst's activity through the apparent activation energy, a metric influenced by various factors working in tandem to catalyze the reaction. As an illustration, Junwang Tang et al. reported the use of Au60s/TiO₂ for the photocatalytic oxidative coupling of methane, achieving an impressively low apparent activation energy of 5.47 kJ mol⁻¹, although methane is recognized to be more inert than ethane due to higher C–H bond energy. (*Nat. Energy* 2023, 8, 1013–1022). Therefore, we consider the apparent activation energy of 18.4 kJ mol⁻¹ for the photocatalytic ODHE on PdZn-ZnO to be reasonable.

We add a corresponding discussion in line 181, “...and increased oxygen consumption due to ethane over oxidation”

Fig. R5. (a) Temperature-dependent photocatalytic ODHE performance. Light intensity = 55.9 mW cm⁻². (b) Temperature-dependent ODHE performance in the absence of light.

5) Should the label of Fig. 3c(10-30 mins) be addressed as C₂H₆+¹⁶O₂?

Response: Yes. The label of Fig. 3c (10-30 mins) is addressed as C₂H₆+¹⁶O₂.

6) In Fig.5, the energy of the dehydration process to form the O vacancy perhaps seems higher than the deoxidation from the PdZn alloy active site. Maybe the energy barriers should be compared from the calculation results. Or the configuration diagram of this two intermediates should be redrawn.

Response: As depicted in the calculated results presented in Supplementary Fig. 38, the energy barrier of deoxidation from the active site of PdZn is 0.92 eV, which is higher than that of the dehydration process to form the O vacancy (0.71 eV). Combining with the lattice oxygen extraction-replenishment cycle experiment over PdZn-ZnO by time-resolved online mass spectrometry (Fig. 3c), the dehydration process to form the O vacancy is more favorable than the deoxidation pathway.

Reviewer #2: *In this manuscript, the authors reported a novel ethylene production pathway through photocatalytic oxidative dehydrogenation of ethane with oxygen under mild conditions, showing great development potential compared to the existing naphtha cracking route with high energy consumption and greenhouse gas emissions. In addition, the mechanism of photocatalytic oxidative dehydrogenation of ethane was studied, and a light-driven lattice oxygen-mediated activation mechanism of ethane was proposed. This work will provide a reliable guide for studying photocatalytic high-value-added conversion and activation mechanisms for alkanes. I think this manuscript can be accepted by Nature Communications after some minor revisions. Authors should carefully consider the following comments and suggestions.*

Response: Thank you for your professional comments. We have responded to the comments point by point and made some changes accordingly in the revised manuscript.

1) *In Fig. 2f, how does the selectivity of ethylene change with temperature? The decrease in the conversion rate of ethane at high temperatures may also be due to the excessive oxidation of ethylene at high temperatures and the consumption of oxygen.*

Response: Under light irradiation, as the temperature rises (below 180 °C), the production rates of ethylene and carbon dioxide exhibit a gradual increase, while the selectivity of ethylene remains approximately at 80%. However, when the reaction temperature surpasses 180 °C, the production rate of carbon dioxide experiences a rapid escalation, and the ethylene selectivity decreases, signaling the onset of ethylene overoxidation. This overoxidation of ethylene is likely the primary factor contributing to the decline in the ethane reaction rate, as it involves substantial oxygen consumption. (Supplementary Fig. 28 and Fig. R6).

We add a corresponding discussion in line 181, "...and increased oxygen consumption due to ethane over oxidation".

Fig. R6. (a) Temperature-dependent photocatalytic ODHE performance. Light intensity = 55.9 mW cm⁻². (b) Temperature-dependent ODHE performance in the absence of light.

2) The charge transfer behavior between PdZn nanoparticles and ZnO in PdZn-ZnO samples may be experimentally supported by X-ray photoelectron spectroscopy. I suggest that the author test the XPS profile of the sample to supplement it.

Response: As depicted in Supplementary Fig. 35 and Fig. R7, the peaks at 341.4 eV and 336.1 eV in Pd-Zn₅(CO₃)₂(OH)₆ are assigned to Pd in the +2 oxidation state, potentially in the form of hydroxides or carbonate. In the XPS spectrum of Pd 3d in Pd-ZnO, two distinct and intense peaks corresponding to Pd⁰ emerge at 334.5 and 339.8 eV. The Pd 3d signal of PdZn-ZnO is positioned at 340.5 and 335.2 eV, precisely between those of Pd-ZnO and Pd-Zn₅(CO₃)₂(OH)₆. This placement reveals the moderate electron deficiency of Pd in PdZn-ZnO. Such electron deficiency may be associated with electron transfer from Pd to the support, indicative of metal-support interactions.

We add a corresponding discussion in line 278, “The higher binding energy of the Pd 3d...”.

Fig. R7. Pd 3d XPS spectra of PdZn-ZnO, Pd-ZnO, and Pd-Zn₅(CO₃)₂(OH)₆.

3) Why is the calcination temperature of the Pd-Zn₅(CO₃)₂(OH)₆ precursor selected at 300°C? Have the authors tried to treat the precursors at other temperatures?

Response: We evaluated the activity of photocatalysts subjected to different calcination temperatures (Supplementary Fig. 12 and Fig. R8). Pd-Zn₅(CO₃)₂(OH)₆ without calcination displayed no photocatalytic ODHE performance as it could not generate photogenerated carriers under light irradiation. The samples calcined at 200 °C exhibited very poor photocatalytic ODHE performance, likely due to incomplete decomposition of the precursors at this temperature. Similarly, samples calcined at 400 °C and 500 °C also demonstrated subpar photocatalytic ODHE performance, attributed to catalyst sintering (Supplementary Fig. 13 and Fig. R9). The samples calcined at 200 °C still retained the nanosheet morphology of the precursor, while those calcined at 400 °C and 500 °C exhibited noticeable agglomeration and sintering of ZnO nanoparticles.

We add a corresponding discussion in line 121, “The calcination temperature of the precursor...”.

Fig. R8. Photocatalytic ODHE performance with Pd-Zn₅(CO₃)₂(OH)₆ calcinated at different temperatures.

Fig. R9. TEM images for Pd-Zn₅(CO₃)₂(OH)₆ calcinated at different temperatures, (a) 200 °C, (b) 400 °C, (c) 500 °C.

4) *Can the precursor decompose completely at 300 °C? I suggest that the author supplement the thermogravimetric analysis test of the sample to dispel my doubts.*

Response: The Pd-Zn₅(CO₃)₂(OH)₆ precursor decomposed rapidly between 200 °C and 300 °C (Fig. R10), indicating that it could be completely decomposed after being held at 300 °C for 4 h.

Fig. R10. Thermogravimetric curves for Pd-Zn₅(CO₃)₂(OH)₆. TG: thermogravimetry, DTG: differential thermogravimetry.

5) Both photogenerated electrons and holes seem to play important roles in the photocatalytic oxidative dehydrogenation of ethane. The authors can try to eliminate photogenerated electrons or photogenerated holes with trapping agents such as AgNO₃ and CH₃OH to investigate their catalytic performance.

Response: Thank you for your advice. With the introduction of hole trapping agents (Na₂C₂O₄ and CH₃OH) and electron trapping agents (NaIO₃, K₂Cr₂O₇ and AgNO₃), the ethylene production rate decreased significantly (Fig. R11). This observation confirms that the photocatalytic ODHE is driven by the synergistic effect of photogenerated electrons and holes.

Fig. R11. Photocatalytic ODHE over PdZn-ZnO using different trapping agents.

Reviewer #3: *Low-temperature oxidative dehydrogenation of C₂H₆ is an appealing approach to the synthesis of C₂H₄ but remains a challenge. In the manuscript, authors presented a PdZn-ZnO photocatalyst for C₂H₆ dehydrogenation to C₂H₄ by O₂ with high activity and selectivity, showing a remarkable superior to the currently-reported thermocatalytic system. Indeed, it is an interesting work. However, this work lacks more detailed experimental evidences and characterizations and theoretical calculations to reveal the reaction mechanism. Hence, authors should address the following points before publication. The more detailed comments as follows:*

Response: Thank you for your professional comments. We have responded to the comments point by point and made some changes accordingly in the revised manuscript.

1) Page 2, line, the optimized C₂H₆ conversion rate enables reach 20%, in which the conversion rate is calculated based on products or C₂H₆ before and after reaction? The carbon balance should be included in this work.

Response: The conversion rate is calculated based on products according to the following equation.

$$\text{C}_2\text{H}_6 \text{ conversion}(\%) = \frac{n(\text{C}_2\text{H}_4) + \frac{1}{2} \times [n(\text{CH}_4) + n(\text{CO}_2)]}{n_0(\text{C}_2\text{H}_6)} \times 100\%$$

where $n(\text{C}_2\text{H}_4)$, $n(\text{CH}_4)$ and $n(\text{CO}_2)$ represent the moles of C₂H₄, CH₄ and CO₂ at the outlet, respectively. $n_0(\text{C}_2\text{H}_6)$ represents the moles of C₂H₆ at the inlet.

The corresponding carbon balance is around 100% (Supplementary Fig. 21 and Fig. R12). The carbon balance is calculated according to the following equation.

$$\text{Carbon balance}(\%) = \frac{n(\text{C}_2\text{H}_4) + \frac{1}{2} \times [n(\text{CH}_4) + n(\text{CO}_2)] + n_0(\text{C}_2\text{H}_6)}{n_1(\text{C}_2\text{H}_6)} \times 100\%$$

where $n(\text{C}_2\text{H}_4)$, $n(\text{CH}_4)$ and $n(\text{CO}_2)$ represent the moles of C₂H₄, CH₄ and CO₂ at the outlet, respectively. $n_0(\text{C}_2\text{H}_6)$ and $n_1(\text{C}_2\text{H}_6)$ represent the moles of C₂H₆ at the inlet and outlet, respectively.

We add a corresponding discussion in line 156, “Moreover, the carbon balance about 100 %...”. We also have given the calculating equations for C₂H₄ selectivity, C₂H₆ conversion and carbon balance in the experimental section, line 428.

Fig. R12. Carbon balance for stability test in Fig. 2c.

2) *The specific role of Pd and Zn during the C₂H₆ dehydrogenation?*

Response: In photocatalytic ODHE, ZnO play a crucial role in activating ethane C–H bonds through photogenerated O⁻ active species. This assertion is supported by in situ electron paramagnetic resonance spectroscopy (Fig. 3b) and corroborating literature (*Nat. Catal.* 4, 2021, 1032–1042). For PdZn nanoparticle, it enhances the migration of photogenerated carriers, leading to the generation of O⁻ reactive oxygen species, thereby facilitating the dehydrogenation of ethane. Through time-resolved online mass spectrometry (Fig. 3c), we observed a lattice oxygen extraction-replenishment cycle in photocatalytic ODHE, with PdZn intermetallic compound particles proving more effective in facilitating this cycle compared to Pd nanoparticles. Additionally, density functional theory calculations affirm a stronger metal-support electron transfer behavior between PdZn-ZnO, further promoting ethane activation and lattice oxygen cycling at the interface.

3) *The enhanced activity is attributed to photocatalysis or photo-assisted thermocatalysis or synthetic effect of photo-thermal synergy? The specific role of photo and thermal? How the photogenerated electron and hole affect the reaction route?*

Response: We attribute the enhanced catalytic performance to photocatalysis for several reasons. Firstly, in the absence of light, negligible product formation is detected below 300 °C, and even at elevated temperatures, only a substantial amount of carbon dioxide is observed (Supplementary Fig. 28). However, during photocatalytic ODHE at 136.6 °C, ethylene is the primary product, indicating that PdZn-ZnO follows a markedly different reaction pathway under dark and light conditions. Secondly, we observed a linear relationship between the ethylene production rate and the light

intensity (Supplementary Fig. 26 and Fig. R13). The wavelength-dependent apparent quantum efficiency (Fig. 2g) also closely aligns with the light absorption characteristics of ZnO at the corresponding wavelengths, confirming that the enhanced catalytic activity can be attributed to photocatalysis.

While light plays a predominant role in photocatalyzed ODHE, we have shown that the reaction temperature also influences the activity of photocatalytic ODHE. In the presence of external heating supply (Supplementary Fig. 28 and Fig. R14), as the temperature rises, the ethylene production rate gradually increases, while the selectivity remains essentially unchanged. However, when reaction temperature exceeds 250 °C, ethane tends to be overoxidized to CO₂ and results in a reduced ethylene selectivity. This indicates that maintaining an appropriate temperature in the low-temperature range can facilitate the evolution of intermediate species.

Photogenerated holes (O^{•+}) have the capability to activate the C–H bond of ethane, as demonstrated in the manuscript through in situ electron paramagnetic resonance spectroscopy (Fig. 3b). This process aligns with the reported activation of C–H bonds in methane by photogenerated O^{•+} species on ZnO (*Nat. Energy* 2023, 8, 1013–1022; *Nat. Catal.* 2020, 3, 148–153; *Nat. Catal.* 2021, 4, 1032–1042). The activation of C–H bonds with O^{•+} generates hydroxyl species on the catalyst surface, a presence confirmed through in situ infrared spectroscopy (Fig. 3e). Subsequently, time-resolved online mass spectrometry detected water generated by the dehydration of surface hydroxyl groups, leaving an oxygen vacancy on the catalyst surface. Photogenerated electrons, on the other hand, activate oxygen and facilitate oxygen dissociation to fill the oxygen vacancy, a confirmation supported by isotopic analysis (Fig. 3c).

We add a corresponding discussion in line 174, “...and showed a linear relationship with the light intensity” and line 181, “...and oxygen consumption due to ethane over oxidation”.

Fig. R13. C₂H₄ production rate under different light intensities.

Fig. R14. Temperature-dependent photocatalytic ODHE performance. Light intensity = 55.9 mW cm⁻².

4) The radical such as ($\cdot O_2^-$) should be detected during the reaction for excluding the radical mechanism.

Response: The detection of the superoxide radical was carried out using methanol as a solvent and 5,5-Dimethyl-1-Pyrroline N-Oxide (DMPO) as a spin-trapping agent by in situ electron paramagnetic resonance (EPR) spectroscopy. No signals were detected in either the dark or light conditions, eliminating the possibility of interference caused by DMPO photolysis (Fig. R15a). We detected the signals of $\cdot O_2^-$ formed via $e^- + O_2 \rightarrow \cdot O_2^-$ over PdZn-ZnO photocatalyst under light irradiation (Fig. R15b). Result suggests that oxygen may have undergone the following evolutionary pathways: $O_2 \leftrightarrow O_2^- \leftrightarrow O_2^{2-} \leftrightarrow 2O^- \leftrightarrow 2O^{2-}$ (lattice oxygen) (*Chem. Soc. Rev.* 2021, 50, 4564–4605). However, no alkyl radicals were detected for PdZn-ZnO in an aqueous DMPO solution saturated with ethane under light irradiation (Fig. R16). Therefore, we think that the photocatalytic ODHE is carried out on the catalyst surface rather than through radical intermediates mechanism.

Fig. R15. (a) In situ EPR spectra in DMPO-containing methanol. (b) In situ EPR spectra for PdZn-ZnO with DMPO in methanol.

Fig. R16. EPR spectra of spin trapped radicals formed during irradiation of aqueous PdZn-ZnO dispersions saturated with C_2H_6 .

5) *The catalyst can drive the reaction under light at 136 °C. How about at room temperature? The specific role of thermal?*

Response: Light inevitably increases the surface temperature of the catalyst and it is difficult to eliminate the photothermal effect during photocatalytic reactions. We try to reduce the temperature of the catalyst by reducing the light intensity. Under 50 mW cm^{-2} light irradiation, the surface temperature of the catalyst is about $32.8 \text{ }^\circ\text{C}$, close to room temperature, and the ethylene production rate is $3.1 \text{ mmol g}^{-1} \text{ h}^{-1}$ (Supplementary Fig. 26 and Fig. R17).

In the presence of external heating supply (Supplementary Fig. 28 and Fig. R14), as the temperature rises, the ethylene production rate gradually increases, while the selectivity remains essentially unchanged. This is in agreement with previous studies

that thermal can promote the evolution of intermediate species under light irradiation (*Chem. Soc. Rev.*, 2021, 50, 2173). However, when reaction temperature exceeds 250 °C, ethane tends to be overoxidized to CO₂ and results in a reduced ethylene selectivity. This indicates that maintaining an appropriate temperature in the low-temperature range can facilitate the evolution of intermediate species.

Fig. R17. (a) Temperature curve of catalyst surface under different light intensities. (b) C₂H₄ production rate under different light intensities.

6). The MS signals of H₂ should be provided during the in-situ C₂H₆ dehydrogenation process detected by MS.

Response: We monitored the hydrogen signal (m/z=2) in the photocatalytic ODHE using online mass spectrum and no hydrogen production was detected under light conditions (Fig. R18).

Fig. R18. Online mass spectrum signals for H₂, H₂O, O₂, and C₂H₆.

7) In Fig. 2c, compared with CH₄ and CO₂, the C₂H₄ productivity fluctuates

significantly while the C_2H_4 selectivity remains well. Please explain this phenomenon.

Response: The activity of photocatalytic oxidative dehydrogenation of ethane depends mainly on the surface active lattice oxygen on the catalyst. In the initial stage, there is a large amount of easily extracted lattice oxygen atoms on the catalyst surface. As the reaction progresses, lattice oxygen atoms that cannot complete the extraction-replenishment cycle are gradually deactivated, leading to fluctuations in the ethylene production rate. On the other hand, the ethylene selectivity mainly depends on the type of metal and the metal-support interaction, so it does not show obvious fluctuations.

REVIEWERS' COMMENTS

Reviewer #1 (Remarks to the Author):

I think this revised manuscript can be accepted.

Reviewer #2 (Remarks to the Author):

The comments have been fully and positively responded to, and the manuscript has been modified accordingly. I strongly suggest its publication in Nature Communications.

Reviewer #3 (Remarks to the Author):

The revised manuscript has already meet the requirements for publishing. Please publish it.

Response to the reviewers' comments

Reviewer #1: *I think this revised manuscript can be accepted.*

Response: We thank the reviewer for the positive comment.

Reviewer #2: *The comments have been fully and positively responded to, and the manuscript has been modified accordingly. I strongly suggest its publication in Nature Communications.*

Response: Thank you very much for your approval and guidance on our manuscript.

Reviewer #3: *The revised manuscript has already meet the requirements for publishing. Please publish it.*

Response: We express our gratitude to the reviewer for acknowledging the research value and providing constructive comments to enhance the quality of this work.